# Review on Starter Pellets: Inert and Functional Cores

**DOI:** 10.3390/pharmaceutics14061299

**Published:** 2022-06-18

**Authors:** Nikolett Kállai-Szabó, Miléna Lengyel, Dóra Farkas, Ádám Tibor Barna, Christian Fleck, Bálint Basa, István Antal

**Affiliations:** Department of Pharmaceutics, Semmelweis University, Hőgyes E. Street 7–9, 1092 Budapest, Hungary; kallai.nikolett@pharma.semmelweis-univ.hu (N.K.-S.); lengyel.milena@pharma.semmelweis-univ.hu (M.L.); farkas.dora1@pharma.semmelweis-univ.hu (D.F.); barna.adam@pharma.semmelweis-univ.hu (Á.T.B.); christian.petszulat@stud.semmelweis.hu (C.F.); basa.balint@pharma.semmelweis-univ.hu (B.B.)

**Keywords:** inert core, drug layering, starter core, pellets, functional pellet core, MCC, isomalt, tartaric acid core, sugar spheres

## Abstract

A significant proportion of pharmaceuticals are now considered multiparticulate systems. Modified-release drug delivery formulations can be designed with engineering precision, and patient-centric dosing can be accomplished relatively easily using multi-unit systems. In many cases, Multiple-Unit Pellet Systems (MUPS) are formulated on the basis of a neutral excipient core which may carry the layered drug surrounded also by functional coating. In the present summary, commonly used starter pellets are presented. The manuscript describes the main properties of the various nuclei related to their micro- and macrostructure. In the case of layered pellets formed based on different inert pellet cores, the drug release mechanism can be expected in detail. Finally, the authors would like to prove the industrial significance of inert cores by presenting some of the commercially available formulations.

## 1. Introduction

The application of ready-to-use cores for the formation of pharmaceutical pellets as starting materials is increasing in the pharmaceutical industry. The development of a relatively easily adaptable technology is offered as an alternative to agglomeration pelletization processes [1,2].

The importance of inert cores is shown by the fact that the first approved modified-release (MR) formulation was a multiparticulate system, a layered pellet containing capsule, manufactured by Smith, Kline & French (now GlaxoSmithKline, Brentford, UK). They used sugar spheres as an excipient to form a rounded shape. The production and commercialization of capsules containing Dexedrine began in 1952, and after a minor color modification, it is still marketed in the United States [3,4,5].

In recent decades, sugar spheres have continued to be popular in the pharmaceutical industry. Due to the development of industrial technologies (equipment, processes) and the use of new excipients, layered pellets are now available not only in capsule dosage form, but also in suspensions, compressed form, or even in the form of orodispersible tablets [6,7,8].

Furthermore, the importance of inert pellet cores is shown by the fact that starter pellet cores made of various excipients are likewise available commercially to the pharmaceutical industry. Thanks to a sophisticated combination of manufacturing and modern analytical techniques (like image analysis), excipient manufacturers now offer even personalized, tailor-made, reproducible particle size distributions (PSD) to formulate and produce drug-loaded pharmaceutical pellets.

In this review, the authors focus on collecting and presenting some frequently used starter cores. The manuscript describes the main properties of the various nuclei, in particular the mechanism of drug release. Finally, the authors would like to prove the industrial significance of inert cores by presenting some of the commercially available formulations.

## 2. Pellets

The pellets, containing active ingredient(s), can be used to build multiparticulate systems, also known as multi-unit systems. The presence of numerous small pellets within a dosage form provides several advantageous technological, physiological, and therapeutic properties for a multi-unit formulation over single-unit formulations (e.g., matrix tablets), making it easy to develop a patient-centered medication with relatively simple industrial methods [9,10,11].

Different from the granules, pellets have an almost perfect spherical shape, compact structure, and relatively smooth surface. Their particle size distribution can be classified into a homogeneous, narrow range [12,13]. Although many pelletizing processes have long been feasible in the pharmaceutical industry, they have been widespread since the late 1970s [14]. There are several methods for the production of pellets, of which mainly extrusion/spheronization, direct pelletizing, powder layering, and solution or suspension layering are used in the pharmaceutical industry [15]. The two main types of pellet structures are the matrix pellets, and pellets with layered structures (Figure 1).

Inside the matrix pellets, the active ingredient and the excipients make a homogeneous system. In a heterogeneously distributed structure, the inert core is seen inside the pellet and it is surrounded by the drug layer. [16]. When comparing the structure, higher drug content can be achieved for the matrix pellets, it is more difficult to obtain the spherical shape. Some methods are also not able to produce the desired narrow size range, for example with direct pelleting). It affects the surface size and consequently influences the coating too. With layered pellets generally, lower drug content can be ensured, although in the case of smaller inert cores, up to 75% *w*/*w* pharmacon content can be reached [2].

However, it is important to note for the future that there are efforts to produce pellets with 3D printing [17]. With this novel technology spherical miniprintlets with various structures (such as the Janus face structure) can be prepared.

Nowadays several starter pellet cores are available to the pharmaceutical developer/manufacturer for the formulation of pharmaceutical pellets. There are many names in the international literature for these particles, for example, neutral or inert pellets or cores. The terms beads, spheres, seeds, nonpareils, starter core, and starter spheres may also be used [16]. There are numerous methods for the production of neutral cores. It can be accomplished in a fluidized bed apparatus or a centrifugal granulator, by an extrusion/spheronization process, and the conventional coating pan methods are likewise available [1,18]. The main difference from the matrix pellets is that no active substance can be used for the production of inert pellet cores, only excipient(s) build up the spherical particle. Inert pellet cores are made of excipients official in the pharmacopeias, such as sucrose, microcrystalline cellulose (MCC), isomalt, and anhydrous dibasic calcium phosphate base, but the use of lactose, tartaric acid, or even silica-based pellet cores is also possible. Table 1 shows commercially available pellet cores for pharmaceutical use.

## 3. Layering and Coating Processes of Inert Cores

Active ingredient-containing pellets can be formed from the inert core by solution, suspension, or dry powder layering process. Each technique has its advantages and disadvantages [21]. During formulation, the most appropriate one should be selected with the properties of the pharmacon and the final product kept in mind.

In the case of the solution or suspension layering process (Figure 2), the active pharmaceutical ingredient (API) is dissolved or suspended in the binder solution [22]. These methods provide a very uniform and smooth surface, however, it must be taken into account that large amounts of solvent/dispersing agents must be added, which can result in a prolonged process, possible change in the solid state of the drug, or poor pellet morphology due to the presence of a solvent [23,24]. The drug-containing liquids (solution/suspension) sprayed onto the seeds are most often aqueous systems.

During powder layering (Figure 3), the surface of the inert cores is first treated with the binder. Thereafter, the drug-containing layer(s) can be applied by continuously adding the drug powder and spraying/drying the binder solution [25,26,27,28]. Powder layering can be particularly advantageous for moisture-sensitive active ingredients [22]. It is important to find a balance between the addition of the powder and the binder [14]. Less liquid is applied during powder layering, so the production time can be reduced, but pellets are formed with a much rougher and more uneven surface compared to drug-loaded pellets prepared by a solution/suspension technique. Generally, in cases where the active ingredient is not stable in solution, or stratification from solution or suspension is rather time-consuming, the use of the powder layering process instead of the solution or suspension layering process is advisable [29].

Both on a laboratory and industrial scale, fluid bed apparatus (bottom spraying with or without Wuster column/Rotofluid) and drum coaters are most commonly used for layering active ingredients onto the surface of the starter cores [30]. With these apparatus, the above-mentioned layering techniques are now considered well-controlled pelletization methods.

The layer formed on the surface of an inert pellet core may contain not only small-molecule active ingredients but also macromolecules or plant extracts. Sugar spheres were used to coat bioactive polypeptide (interleukin-11) as an excipient to produce a delayed-release formulation [31]. Tyagi et al. used sugar spheres likewise to develop multiparticulate systems with targeted drug release containing peptides [32]. Benelli et al. layered polyphenol-rich *Rosmarinus officinalis* extract onto the surface of MCC spheres [33]. In a fluid bed apparatus, MCC-based spheres as inert cores were used for layering *Petroselinum crispum* extract [34]. After the layering process, the pellets are most often coated with film-forming polymers. There are many reasons for forming a polymer coating layer. The purpose of the process may be to mask the taste, protect the drug from external effects (e.g., moisture), or, last but not least, to modify the release of the active ingredient from the pellets. During the film coating process, the thickness and morphology of the resulting coating are key factors that can ultimately affect the quality and stability of the formulation [35]. The thickness and uniformity of the real polymer film coat formed during the process are affected by several factors, such as operating conditions, properties of particles, and applied coating dispersion [36].

Examples of process parameters are inlet air temperature [37] and humidity [38]; spray rate, atomization air pressure [39], or curing time and temperature [40] may affect the quality of the coating layer.

Material attributes include, for example, knowledge of the mechanical properties of the pellets [41] or surface roughness, which may also affect the quality of the polymer coating.

There are many methods for examining the real thickness and uniformity of a film coat, including destructive (SEM) and non-destructive (Raman or NIR spectroscopy, terahertz pulsed imaging, dynamic imaging analysis, and optical coherence tomography) techniques [35].

If the drug and the film-forming polymer are incompatible, a seal coat must be formed between the active substance-containing layer and the film coating. This can be observed in the case of several commercially available multiparticulate dosage forms (capsules, tablets) containing proton pump inhibitors (PPIs). The inert core can be directly surrounded by the API-bearing layer. The film coating contains polymers, that are predominantly acidic molecules and insoluble at low pH (pH 1–3) but soluble in the small intestinal medium. During the coating process, or even during storage, the structure of the acid-sensitive PPIs may be compromised due to chemical interactions. To prevent this, several studies have suggested the preparation of a seal coat between the active ingredient layer and the gastro-resistant film coating, which is used by the vast majority of products on the market today [42,43,44]. The seal coat is formed by coating with rapidly dissolving film-forming polymers. A further advantage of forming a protective layer is that the smoother surface decreases the prevalence of film formation defects, which is extremely advantageous for the next, i.e., gastro-resistant, intestinal-soluble coating [45].

The drug-loaded, polymer-coated pellets can be used to build up multiparticulate single-unit systems (capsules, tablets)—some examples are presented in the next chapter. Another option is, to measure the accurate dose of small particulates, which is an innovative drug delivery device, specifically designed for dosing and orally dispersing microspheres/pellets. One such device is the Sympfiny^TM^ oral syringe, which is available in two sizes (1 mL and 2 mL). It fits the standard bottles and can control and ensure the precise dosage of layered pellets [46,47].

Some unique compositions and structures have been developed for layered pellets. The first layer applied directly on the surface of the inert core does not necessarily contain an active ingredient, such as in the case of the Instaspheres TA (Ideal Cures, Mumbai, India) spheres. They are ready-to-use seal-coated tartaric acid pellets, containing only excipients (binder polymers, solvents, plasticizers, adhesion-controlling excipients) as a seal coat. The system is a suitable pH-modifying functional starter core for weakly basic drugs. Due to the seal coat, the active ingredient does not contact directly with the tartaric acid, and also the retention of the tartaric acid is necessary for the prolonged release of weakly basic pharmacons [48].

Zhang et al. prepared a gastro retentive delivery system with a hollow structure. As the first step of formulation, sugar spheres were coated with a polymer (Surelease: Eudragit NE). Afterward, soaking the coated core in water, the sucrose left the system, leaving the shell behind, followed by further coating steps after lyophilization [49]. In the special case, when the core has a porous inner structure, the drug can be loaded with a solvent evaporation technique [50,51].

## 4. Characterization of Starter Cores, Layered Pellets—Particle Size, Shape, Surface

The pharmaceutical industry often uses measurements of morphological descriptors and micro-/macrostructural properties to evaluate the quality characteristics of solid dosage forms. Understanding the effect of process and material parameters on the characteristics of the final product is crucial in the production of safe and effective drug formulations with consistent quality. This approach is known as QbD (quality by design) [52].

The critical quality attributes are chemical, physical, biological, or microbiological characteristics that have a significant (critical) effect on the desired quality of a preparation (in particular efficacy and safety, mainly summarized in the Quality Target Product Profile). They must be within a certain limit or range, or they may need to follow an ideal distribution to ensure the prescribed and expected (defined) quality of the medicinal product [53]. Table 2 shows a few properties of inert pellet cores that can be critical quality attributes in some cases.

The most important pharmaceutical technology requirements and quality characteristics for neutral cores and layered pellets are particle size, particle size distribution, appropriate particle shape (spherical), and surface/roughness. Determining the flow properties (flowability) of the pellet (starter, drug-loaded, film-coated) is also important. In the case of porous particles, in addition to knowing the tapped and bulk density, the determination of true density (the quotient of the weight of the pellet and the solid volume without pores) is likewise important. Further, moisture content/hygroscopicity, negligible friability value, and sufficient hardness are important properties. In the case of starter cores, the manufacturers carry out even more tests, like heavy metal contamination or microbiological test data. The properties detailed above are extremely important for the processability and a well-functioning final dosage form [2]. Of these properties, size, shape, and surface are discussed in more detail in this manuscript.

*Particle size:* For most inert pellet cores, several particle size fractions are available within the size range of 200 μm to 2 mm, which can be characterized by a very narrow width of particle size distribution. One reason for the existence of so many fractions is that they allow the development of numerous formulation variations and dosage strengths while using excipients with the same chemical structure. In the literature spherical particles with a particle size smaller than 500 μm are also referred to as micropellets [61] (Figure 4).

The particle size fractions of commercially available various types of nuclei are shown in Figure 5, and several excipient manufacturers produce inert cores with such a tiny particle size.

Small particle size is preferred when high drug levels (up to 75% *w*/*w*) are required due to the dose of the pharmacon [2]. Furthermore, in the case of an unpleasant-tasting active ingredient, the taste can be masked with simple cost-effective industrial solutions.

Micropellet-containing suspensions are of outstanding importance in pediatrics, they can even be used in the therapy of neonates [61]. In the formulation of an active ingredient, inert cores characterized by a larger particle size are generally used as excipients to carry a lower amount of active ingredient.

Several factors must be considered when preparing pellet-containing tablets (modified-release or orodispersible tablets). In addition to the excipient factors (e.g., cushioning agents, excipient-pellet ratio) and tableting parameters (compression force, compression speed), the size of the API-loaded pellet must be appropriate [62,63,64,65,66]. The ideal pellet size for tableting, including the active ingredient and the film coating is considered to be a fraction between 500 and 800 μm (Figure 4). Larger inert cores are also suitable for pellet-filled capsules (gastro-resistant or sustained-release) [55]. The applicable size limit for sprinkle preparations, based on the masticated food particles, is defined in the FDA Guidance for Industry. It sets the maximum limit of bead sizes: they should not exceed 2.5–2.8 mm [67,68]. This size limit is especially important in the case of enteric-coated pellets, where unwanted chewing can compromise the effectiveness of the API.

**Figure 5 pharmaceutics-14-01299-f005:**
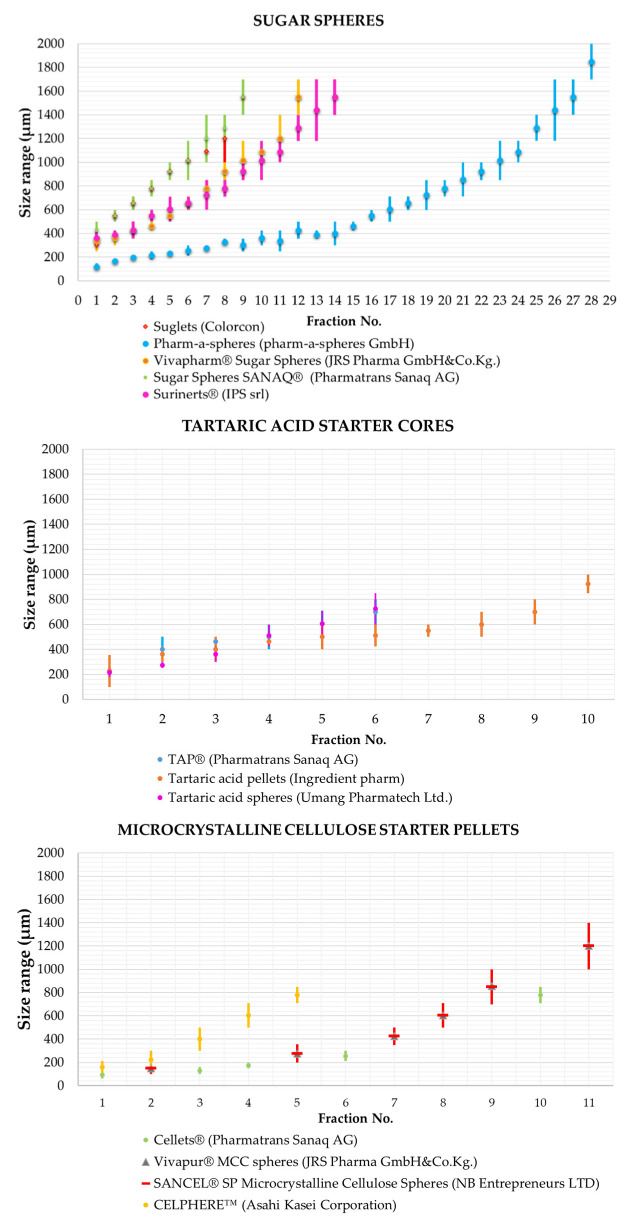
Examples of pellet size ranges available on the market: sugar spheres [69,70,71,72,73], tartaric acid starter cores [74,75,76], and microcrystalline cellulose starter pellets [77,78,79,80].

*Pellet shape:* Shape is a critical quality factor, as a rounded/spherical shape is preferred. Figure 6 shows scanning electron micrographs of various inert pellet cores showing that the investigated, marketed particles are nearly spherical.

Several shape factors are commonly used in the literature to characterize the nearly spherical shape of pellets [81,82,83]. The most widely used ones are roundness, circularity (*C*), aspect ratio (AR), and three-dimensional form factor (e_c3_) [84]. In the case of pellets, it is necessary to quantify the deviation of the projection image from the circular shape. The factor of circularity is suitable, which can be calculated as follows:(1)$$C=\frac{4\times \pi \times A_{p}}{p^{2}}$$

Or inverse
(2)$$\frac{1}{C}=\frac{p^{2}}{4\times \pi \times A_{p}}$$
where *C* is the circularity, *A_p_* is the area of the projection of the grain, and *p* is the perimeter.

Aspect ratio (AR) is a general measure, the ratio of the major and minor axis (or maximum and minimum diameters) of a particle. The value of AR, as well as the other shape descriptor parameters, depends on the type of image analysis (Figure 7) program used, or more correctly, which type of diameter the software takes into account for the calculation [85].

The value of the above-mentioned shape parameters for a particle with a perfectly spherical shape is 1. The degree of deviation from this numerical value indicates the deviation from the ideal shape [86]. According to the literature, the production process for pellets is considered to be appropriate if the value is between AR 1.0 and 1.2 [87].

It is important to note that roundness and AR value are widely used for routine measurements, although they provide information about the projected image, as previously mentioned. Sphericity (*ψ*) provides information about the shape of a particle in three dimensions. Wadell originally defined this property as the true sphericity of a particle, which is the ratio of the nominal surface area (*S_N_* is the area of a sphere whose volume is equal to the volume of the particle) to the actual surface area (*S*) of the particle [88], and can be calculated as follows:(3)$$\psi =\frac{s_{N}}{S}$$

In reality, however, determining the actual surface area of a particle has been an extremely difficult measurement [89]. In recent decades, with the development of technology and the introduction of new testing equipment and software, it has become possible to study the exact and complete 3D morphology of particles of the same size as pellets. Thus, for example, micro-CT (μCT) or synchrotron radiation X-ray computed microtomography (SR-μCT) can also be used to examine the sphericity of pellets [90,91]. The latter method is also applicable for the study of drug distribution of minitablets of the same size range as pellets [92].

*Surface:* In addition to size and shape, surface roughness is a critical parameter for inert cores, as it indirectly determines the specific surface area. An improperly smooth surface can cause variability in the thickness of the coating, which can result in insufficient drug release [93,94].

In the case of the two-dimensional image’s surface structure, roughness can be estimated by determining the center of gravity and the perimeter of the grain contour. The mean value of the measured “radi” is equal to the radius of a perfect sphere, but the surface irregularities result in a decrease in the value, resulting in a smaller circumference of the pellet. Therefore, in the case of a two-dimensional grain, the surface roughness can be determined:(4)$$s_{r}=\frac{2\times \pi \times r_{e}}{P}$$
where *s_r_* = surface roughness, *r_e_* = arithmetic mean of the distances between the center of gravity and the circumference for a given α, *P* = the circumference of the grain outline. The surface roughness of a perfect sphere equals 1, for pellets this value is generally less than 1.

The three-dimensional shape factor combines surface roughness and sphericity:(5)$$e_{c3}=\frac{{\left(\frac{2\times \pi \times r_{e}}{P\times f}\right)}_{1}+{\left(\frac{2\times \pi \times r_{e}}{P\times f}\right)}_{2}}{2}-\sqrt{2-{\left(\frac{b}{l}\right)}^{2}-\left(\frac{t}{l}\right)}$$
where subscripts 1 and 2 indicate the outline of the two measured two-dimensional grains, *l* = the length of the ellipse, *b* = the width of the ellipse, *t* = the thickness of the ellipse, and *f* is a factor that can be applied to correct the surface roughness value as follows [82]:(6)$$f=1.008-0.231\times \left(1-\frac{b}{l}\right)$$

In general, it is recommended to indicate the coefficient *e_c_*_3_ and the surface roughness *s_r_* together, so that the grain shape and the grain surface can be characterized separately. The three-dimensional shape factor, *e_c_*_3_ of a perfect sphere is 1.0, while the shape factor of coarse-grained spheres or non-spherical particles is less than 1.0 [82].

## 5. Types of Starter Cores

It has already been shown in Table 1 that various starter cores are commercially available for the production of pharmaceutical dosage forms. Figure 5 shows that within the type, the manufacturers can still select the most suitable core for their pharmaceutical drug delivery system from a number of size fractions. Table 3 shows the properties of some commonly used pellet cores.

The first inert cores used in the pharmaceutical industry were sugar spheres, as already mentioned in the introduction. They have been used since the 1950s and are still very popular today for the formulation of various dosage forms. This is also due to the fact that sugar spheres are the only starter cores for which not only the raw material but also the pellet itself is monographed in pharmacopeias [95]. As shown in Table 3, sugar spheres are composed of sucrose and maize starch. According to the USP-NF, the starch content is 62.5–91.5% *w*/*w*, and the residual amount is starch [96]. According to Ph. Eur. 10. for sugar spheres, the sucrose content is defined, it should be 92% *w*/*w* or less [97]. The residue component of the core is starch but may contain starch hydrolysates and/or colorants. For multiple reasons, the sucrose and starch content of sugar spheres is of utmost importance from a technological point of view. Manufacturers use water in the manufacturing process of sugar spheres, and the residual moisture content can be either mobile or bound. It can be observed, that the starch content in sugar spheres is related to the amount of bound water. The higher the amount of starch in the sugar spheres, the higher the measured drying loss (LOD) value- due to the quantity of bound water. However, the mobile water content, which can be examined, for example, by estimating water activity, can be considered a critical quality attribute when formulating a moisture-sensitive active ingredient [16]. The solubility of sugar spheres is also important from a technological point of view. As the sucrose that builds up a sugar sphere is freely soluble in water, while starch is practically insoluble in cold water, the solubility of the core depends on the ratio of the components. However, according to the sugar content defined by the pharmacopeias, it can be said that sugar spheres are soluble in aqueous media and have hygroscopic properties that are challenging at the beginning of the drug layering process [98]. To overcome this and its fragile properties, a seal coating is often used prior to drug layering [99]. Marabi et al. studied sugar spheres in water and the solubility of various organic solvents. It takes less than 75 s for a sugar sphere (d_90_ = 706 µm) to dissolve completely in water at 30 °C [100].

Ozturk et al. tested sugar pellets coated with ethylcellulose containing phenylpropanolamine. They found that the mechanism of drug release is mainly affected by the osmotic pressure, and only to a lesser extent by the water-filled pores of the membrane diffusion [101]. According to Lecomte et al., due to the water getting inside the coated sugar pellet during dissolution, the hydrostatic pressure ‘squeezes out’ the sugar solution of the active substance. In case the mechanical properties of the film are not adequate, cracks in the coating can occur [102]. As mentioned earlier, Zhang et al. used sugar spheres to produce hollow-structured particulates to formulate a bioadhesive floating multiparticulate system [49].

Another non-conventional application of sugar spheres is that they are suitable for the preparation of porous titanium scaffolds as a space holder. To produce titanium foams, the titanium powder and sugar spheres were homogenized in appropriate proportions and compressed. Before sintering, the sugar spheres were removed by dissolution to form the porous structure characteristic of solid foams [103].

As an alternative to sugar spheres, numerous inert pellet cores are made of polyalcohols, such as mannitol, xylitol, or isomalt. They are commercially available and water-soluble. The water solubility of the above-mentioned inert cores is of utmost importance in many cases, for example, during the dissolution of poorly water-soluble drugs from the sustained-release pellets [59,104]. As the drug layering process of water-soluble cores is slow, and in some cases, the absorption of aqua into the inert pellet core occurs, starter cores made of other water-insoluble excipients have appeared on the market [105,106].

Among the excipients used for pelletizing processes, Dukić-Ott et al. called microcrystalline cellulose a golden standard [107]. The MCC-containing wet mass has adequate cohesive properties during extrusion/spheronization processes. It can bind large amounts of water, and even the movement of water in the wet mass is controlled, so the phases do not separate during the extrusion process. Due to its sufficiently porous structure and large surface area, the MCC-bearing wet mass also has sufficient plasticity during spheronization. These advantageous properties allow MCC-based matrix pellets and MCC-based starter cores to be produced with high yields. The particulates are characterized by good sphericity, low friability, and smooth surface [108]. Pure MCC starter cores are insoluble in water, which is an advantageous feature when spraying aqueous solutions/suspensions during coating. Compared to layering sugar spheres, the process will be faster when the inert cores are made of MCC. Very specific use of MCC beads was experimentally found successful: Celphere^®^ microcrystalline pellets were used intravenously as a vascular embolization agent [109].

Nevertheless, the disadvantages of using MCC should be taken into account, as shown in several publications. Thus, due to the large surface area and the porous structure, possible drug adsorption may occur [110,111]. The MCC can react chemically with drugs, resulting in incompatibility problems [112,113].

To promote disintegration from MCC-containing matrix pellets and ensure faster, adequate drug release, MCC has been combined with various materials (water-soluble excipients to aid disintegration) during matrix pelletization. Wetting MCC with an alcohol/water mixture can reduce the mechanical strength of the pellet, resulting in faster disintegration and thus faster drug release [114]. There have been several publications on potential excipients that would reduce/replace the amount of MCC in extrusion/spheronization processes for matrix pellets. Liew has articulated some of the important properties that may make an excipient suitable for MCC replacement [115].

Bornhöft et al. studied the applicability of different carrageenan types in pelletizing processes. They found *κ*-carrageenan to be the most suitable for pellet production [116]. Pectin derivates were also studied for the same reason—only the sparingly water-soluble pectic acid resulted in high-yield pellets with a suitable shape [117,118,119]. Several studies report the use of chitosan in pelletization processes [120,121,122]. In addition to the materials described above, several other excipients can be used to reduce/replace the amount of MCC in matrix pellets.

The need of exchanging MCC in inert cores with other excipients while retaining the beneficial properties and reducing its possibly avoidable effects is well known. Starter pellet cores based on dibasic calcium phosphate appeared on the market a few years ago. They contain 20% *w*/*w* MCC and the remaining 80% *w*/*w* is anhydrous calcium dihydrogen phosphate (DCPA). Like the pure MCC pellets, they are not water-soluble, so layering is a similarly simple and fast process [58,105,123].

Pellet cores with a spherical shape suitable for drug layering can be made of water-soluble tartaric acid. The functional starter pellets, consisting of 100% tartaric acid, can also be considered as pH modifiers of the microenvironment for a sustained release formulation. This is the perfect choice for active ingredients that have poor solubility in a higher pH environment, such as in the lower part of the intestinal tract [124]. In the case of the previously mentioned InstaSpheres TA (IdealCures) spheres, according to the manufacturer’s specification, the tartaric acid content is not less than 75% *w*/*w*, the remnant is the film coating [48,125].

## 6. Study of the Behavior of Starter Cores in Aqueous Medium with Microfluidic Device

As mentioned earlier, the behavior of inert pellet cores in an aqueous medium is extremely important. On the one hand, they can affect the time of the layering process with the active ingredient. However, on the other hand, their solubility may also affect the release of the drug from the final dosage form.

The behavior of pellet cores in aqueous media can be observed under a microscope by using a simple microfluidic device (Figure 8 and Figure 9).

The experiments were carried out using a microfluidic device tailor-made via soft lithography. The medium flows with a low flow rate through the micro-scale channels and the pellets are observed under the microscope with photographs taken at pre-set times. The collected samples can be analyzed after certain time points, too.

A similar microfluidic testing device was used for release tests by Amoyav et al. [126] and Ren et al. [127].

The microcrystalline cellulose cores do not show rapid disintegration in aqueous media, and pH dependency cannot be observed. However, due to the porosity of the cores, a light swelling effect is seen. The cores containing 80% *w*/*w* dibasic calcium phosphate and 20% *w*/*w* MCC also disintegrate over a longer period of time. Similar to sugar spheres and isomalt cores, tartaric acid starters dissolve very quickly in aqueous media. Sodium bicarbonate seeds dissolve in an aqueous medium and react in an acidic medium; the formation of carbon dioxide bubbles aids the disintegration, while neutralizing or alkalizing the dissolution medium. This can be advantageous in certain physiological and pathological states [128].

## 7. Dissolution Studies and Release Mechanism from Inert Core-Based Pellets

As mentioned, drug-loaded starter cores usually have additional coating layer(s) that generally modify the drug release. Several commercial formulations contain layered structured pellets coated with enteric polymer. In the case of an enteric coating, due to the properties of the film-forming polymer, there is practically no drug release in an acidic medium. When the carrier system leaves the stomach and enters the intestinal tract, the pH increases, thus the polymer coating can dissolve and release the active ingredient. The latter is illustrated in Figure 10. Compared to water-soluble beads, where the core materials also dissolve, the water-insoluble starter pellets are eliminated unaltered from the body.

Zakowiecki et al. used different types of starter cores, such as water-soluble sugar and isomalt cores, and water-insoluble MCC and DCPA starter core for the preparation of poorly-soluble diclofenac sodium-loaded enteric-coated pellet formulations. Their results showed that the water-insoluble cores resulted in a lower degree of diclofenac release in the acid phase compared to the formulations which are based on water-soluble cores [123].

In many cases, the pellets have permeable film coating. These usually contain polymers that are insoluble in an aqueous medium in the physiological pH range (pH 1–8). The dissolution consists of the following processes. A possible drug release mechanism can be, that the active ingredient diffuses through the intact polymer layer (Figure 11A). When the concentration of plasticizer in the film coating is high and/or its distribution is not uniform, during dissolution, the water-soluble plasticizer molecules form channels in the layer, and the drug is liberated via the channels (Figure 11B). It is also possible, that water-filled pores are formed in the film coating and the active ingredient diffuses through them (Figure 11C). The release of the active ingredient may also be affected by the presence of an osmotic agent in the formulation.

In the case of layered pellets, for example, sugar spheres act as osmotically active pellet cores. During dissolution they generate extra osmotic force, which increases the net water ingress towards the inside of the pellet, thereby accelerating drug release (Figure 11D) [101].

Of course, it cannot be claimed that only one mechanism governs the drug release for a coated pellet. Usually, these processes take place simultaneously, but one mechanism has the most crucial role in the drug release. In addition to permeability and drug solubility, numerous other factors must be considered, which can influence the in vivo release and during dissolution: the condition of the stomach at the time of ingestion, the rate of gastric emptying, and even the physiological pH range of the gastrointestinal tract [129].

Muscher et al. studied the dissolution profile of ethylcellulose-coated pellets of different structures (matrix and layered) that contained APIs with different solubilities in media of various osmolality (0.280–3.580 Osmol/kg). According to the results, the dissolution profile was mainly influenced by the solubility of the active substance. The dissolution profile of the ethylcellulose-coated pellets containing diltiazem hydrochloride, which was characterized by good water solubility, was much faster than of the pellets containing sparingly water-soluble theophylline. There was no difference in drug release between the matrix and layered pellets [130]. The type of the inert core (MCC pellet/sugar sphere) under physiological (0.280 and 0.620 Osmol/kg) conditions had no role in the study of layered pellets containing the active ingredient with good water solubility. However, dissolution studies in hyperosmolar media (1.860 and 3.580 Osmol/kg) show a significant reduction in the dissolution of diltiazem from MCC-based layered pellets, while only slightly decreasing in the case of sugar-based layered pellets [131].

According to our previously published results, in the case of poorly water-soluble diclofenac sodium-containing pellets coated with Eudragit RS and Eudragit RL, the starter core material affected the amount of in vitro dissolved drug even under isosmotic conditions. The shape of the dissolution curves of the coated pellets prepared from water-soluble neutral pellet cores (isomalt pellets, sugar spheres) are similar but differ wildly from the drug release of MCC-based layered pellets. By monitoring the change in particle size during the dissolution of the permeable-coated pellets, it can be seen that the degree of swelling is independent of the neutral pellet core. Also, the cracking of the coating is not essentially involved in drug release. Both isomalt and sugar-based pellets develop significant osmotic pressure during dissolution (Figure 12), which promotes drug release [59].

Inert pellet cores made of water-soluble excipients as well as water-insoluble materials, with patent registration pellet cores, containing different proportions of isomalt and MCC, were prepared [132,133]. For composite pellet cores, the release rate of sodium diclofenac depended on the isomalt content of the starter pellet core for the Eudragit RS coating. Furthermore, the effect of the osmolality of the dissolution medium on drug release depended on the ratio of MCC to isomalt [104].

Pareek et al. studied tartaric acid-based pellet cores, which own a pH-modifying function. In the case of weakly basic drugs, they can provide adequate drug release even in a high pH dissolution medium due to the acidic microenvironment formed within the pellet (Figure 12). For their studies, they used the previously mentioned coated tartaric acid pellets, Instaspheres, which had an extended-release coating in addition to the dipyridamole layer. Comparing the formulation of sugar sphere-based pellets with the Instasphere-based pellets, better drug release was achieved in a high-pH dissolution medium. The cause of this lies in the dissolution of tartaric acid and the resulting acidic microenvironment within the pellet. This was also the case when the surface of the starter sugar sphere was treated with tartaric acid [48].

## 8. Preparations Based on Inert or Functional Pellet Cores and Their Pharmaceutical Aspects

Due to their many advantageous properties, core-based pellets are used in a variety of formulations containing numerous pharmaceutical active ingredients. The patient-centric drug formulation in the case of pediatric preparations requires different tasks: fine-tuning of dose according to age or weight, easy swallowability, better taste, and once-a-day intake. To meet these requirements, various technological solutions were developed, and many of these are based on multiparticulate drug delivery systems. For newborns, infants, and smaller children, liquid dosage forms are preferred to avoid the risk of choking. For children above 6 years, besides liquid formulations, solid dosage forms (tablets, effervescent formulations, orodispersible tablets, films, pellets, or minitablets) could be safely administered [134]. Several studies prove that smaller particles (<3 mm) can be more easily swallowed by children [135,136,137]. This emphasizes the advantageous use of multiparticulate preparations for this group of patients.

For the elderly, dysphagia is a common problem. One in nine older community-dwelling adults have symptoms that amount to dysphagia, which are likely to be under-reported and under-recognized [138]. Patel et al. developed a coated pellet in an easy-to-swallow gel formulation for the sustained release of gliclazide for patients with dysphagia [139].

Oral gels and viscous solutions, syrups, or mucilages are a good choice to ease the intake of minitablets or pellets. The miscibility, sedimentation, and disintegration time of pellets in different vehicles influence the performance [140].

The palatability is influenced by the size of the pellets: rough mouthfeel was found to be more significant with core granules with particle sizes ≥ 200 µm, and less rough mouthfeel was observed with core granules composed of water-soluble additives [141]. A novel study [142] pinned out that besides dimensions and appearance, palatability is a very common problem for the elderly. By the intake, however, breaking tablets into equal halves, sticky coating layers, or the dosing of liquid preparations are all common problems reducing the patient’s compliance. This points out that multiparticulates are relevant to younger and older patients as well. Extended-release multiparticulate formulations also help keep the therapy adherence at a higher level, by allowing once-daily intake of the medications.

Erosive oesophagitis and gastroesophageal reflux disease (GERD) can be healed or maintained by the acid-labile compound, omeprazole, wherein the treatment of pediatric patients both delayed-release oral suspensions and oral capsules can be used. The oral suspension contains enteric-coated particles, where sugar spheres serve as a base component [143].

Table 4 summarizes a few drug preparations available in Hungary, where the API is carried by inert pellet cores, namely sugar spheres. Of course, there are also pharmaceutical preparations in Hungary that have an MCC or other inert core, but it is difficult to find those preparations in the Hungarian databases.

The capsules shown in Figure 13 contain layered pellets. Image analysis was performed, the results of which are also shown in the figure. Based on the AR value, the investigation pellets meet the requirements. Their particle size, as mentioned earlier, is in the larger particle size range than the size of micropellets, which are more suitable for capsule filling. The image in the first column shows a halved esomeprazole-loaded pellet (Emozul, Krka), on which three different coating layers formed on the surface of the inert pellet core, i.e., the active substance-containing layer, the protective layer, and the gastro-resistant, enterosoluble film coating can be recognized [145,146]. The halved pellets (middle column) clearly show the structure of the sugar sphere, the active ingredient layer, and the functional polymer coating [147]. The last column shows the retard formulation containing ascorbic acid. On the surface of the sugar beads, the vitamin C-bearing layers are alternated with layers of neutral film coating, thus forming an onion-like structure. The average bioavailability of ascorbic acid is 98.6% [148,149].

Riomet ER Extended-Release Oral Suspension contains metformin hydrochloride both in the extended-release pellets and diluents. The composition of the oral suspension according to the Summary of Product Characteristics (SmPC) is described as containing the following inactive ingredients: colloidal silicon dioxide, dibutyl sebacate, ethyl cellulose, hypromellose, magnesium stearate, methyl paraben, microcrystalline cellulose, microcrystalline cellulose and carboxymethyl cellulose sodium, propyl paraben, sucralose, strawberry flavor Type FL #28082 (flavoring ingredients, propylene glycol, and glycerin) xanthan gum, and xylitol. As the MCC spheres are not monographed in the pharmacopeias, from the SmPCs the type of the pellets (matrix/layered) is not always obvious if the preparation contains starter pellet cores. In a study, however, the structure layered on the surface of small inert cores is presented [150].

Both Adderall XR and Focalin XR capsules can be opened and sprinkled on food to ease the intake.

Producers offer a wide variety of multiparticulate preparations with smart technological solutions, where the individual subunits or their mixture exhibit different properties [151]. Table 5 shows examples of marketed products of different dosage forms and indications, where inert pellet cores were used for multiparticulate formulations.

Innopran XL extended-release capsule is designed based on the Diffucaps^®^ technology (Adare Pharmaceuticals), where sugar cores serve as a base, and a dual-layer controls the release of the highly lipophilic, thus very well-absorbing, propranolol hydrochloride. This technology ensures that the blood level is aligned to the circadian rhythm after a bedtime intake [168].

The spheroidal oral drug absorption system (SODAS^®^) is a registered formulation of Elan Drug Technologies (merged with Alkermes). This patented technology uses neutral starter cores as the base. The API is layered on the surface and as an external layer, one or more rate-controlling polymer (water-soluble, insoluble, or even pH-responsive) is added to reach a customized drug delivery. These subunits can also be compressed into a tablet or packed into a capsule. A wide variety of active ingredients can be formulated based on SODAS^®^ technology. Amongst others, it is advantageous in the following cases: if the API has a short half-life (several intakes are necessary with conventional formulations), if a breakthrough pain could occur, or if there’s a chance for drug abuse. It is also preferred when fed status can influence the release, or when the patients have dysphagia, as this technology allows once or maximum twice daily intake with minimal PTF (peak-to-trough fluctuations). SODAS^®^ technology is also capable of ensuring pulsatile release, where the natural secretion of different materials can be mimicked. Ritalin LA^®^ (methylphenidate; Novartis) and Focalin™ XR (dexmethylphenidate; Novartis) are used to treat ADHD, and both preparations can provide a bimodal release due to the registered, spheroidal oral drug absorption system (SODAS^®^), where the inert starter cores are covered with different layers [169].

Verelan^®^ PM uses the registered CODAS^®^ (Chronotherapeutic Oral Drug Absorption System) technology, which is designed for bedtime dosing and targets a 4 to 5-h delay in drug delivery. The controlled-onset delivery system results in a maximum plasma concentration (c_max_) of verapamil in the morning hours, 4–5 h after ingestion. These sugar sphere-based pellet-filled capsules provide a controlled, extended-release of the drug in the gastrointestinal tract by the combination of water-soluble, pore-forming, and water-insoluble polymers. After the dissolution of the water-soluble polymer, the insoluble polymer still functions as a barrier and controls the release of the verapamil salt. The rate of release is essentially independent of pH, posture, or food. The pellets can be consumed by being sprinkled on food, for example very soft, room-temperature apple sauce; they have to be swallowed without chewing to preserve the function of the rate-controlling layers. Multiparticulate systems, such as Verelan^®^ PM, were shown to be independent of gastrointestinal motility [170]. Further, several MDDS sprinkle products are designed based on chronotherapeutic oral drug absorption system (CODAS^®^) technologies.

Bylvay^TM^, an orphan medicinal product (EU/3/12/1028 on 17 July 2012) used in the treatment of progressive familial intrahepatic cholestasis, is formulated as oral or sprinkle capsules. The intake is advised with food, for example, sprinkled on apple sauce, which food is usually advised to be applied for the intake of enteric-coated medications [171].

It is not a rare case to combine coated inert pellets with different release patterns to reach a specific effect, a targeted blood level. Metadate CD^®^ (methylphenidate hydrochloride; UCB) is a central nervous system stimulant for the treatment of ADHD. Metadate CD^®^ performs a biphasic release pattern, which is attributed to the IR (immediate release, 30% of the dose) and ER (extended-release, 70% of the dose) beads included in the capsule. Dose titration is possible as Metadate CD^®^ is on the market in different strengths (10, 20, 30, 40, 50, or 60 mg of methylphenidate hydrochloride). Metadate CD^®^ administered as sprinkles on applesauce showed bioequivalent systemic exposure (as c_max_ and AUC) of methylphenidate compared to the intact capsule [158].

Rytary^®^ or Numient^®^ is a composition of carbidopa-levodopa in a hard capsule for the treatment of Parkinson’s disease. It combines immediate-release pellets (1/3) with extended-release components (2/3) to ensure a quick and sustained effect, as well as to reach a blood level without high PTF. As a functional component of the layers, tartaric acid serves as a pH-modifier to increase the rapid onset of action by facilitating the absorption of levodopa even in gastroparesis patients [172].

## 9. Future Perspectives

The use of inert pellet cores has the potential to continue playing a significant role in the pharmaceutical industry in the development of formulations in the coming years. This can be traced back to several reasons. In the case of oral MR formulations, the use of pellets has several advantages over the so-called single unit dosage forms and is therefore still in the spotlight [173].

Furthermore, it is important to note that the aging of the population and the increment in the number of chronic patients contribute to an increase in the number of polymedicated patients. Fixed-dose combination (FDC) formulations containing a minimum of two different active ingredients within a dosage form are particularly attractive adherence enhancers to polymedicated patients. The preparation of FDC formulations from a combination of pellets containing different active ingredients can be easily carried out [174,175]; the pellets can also be prepared by a layering process using inert pellet cores [176]. Innovative patient-centric dosage forms, such as medicated straws with layered pellets [177] can also be easily implemented. Furthermore, the development of 3D printing and the appearance of additional excipients may contribute to the production of new types, structures, or even composite inert cores. Starter cores may also be suitable for carrying solid dispersions or nanoparticles on their surface.

Pioneering research demonstrates the use of functional cores (e.g., calcium carbonate) which can be loaded with a drug in the inner porous structure [50].

## 10. Conclusions

Today various starter pellets are commercially available. The cores have several particle size fractions in a narrow size range prepared for the pharmaceutical industry as ready-to-use excipients for drug layering techniques. The manuscript presents the most often used inert cores in the pharmaceutical industry, along with the process of drug layering. The main properties of inert pellets, which may be decisive in the development of a multiparticulate formulation, where the active ingredient surrounds the surface of an inert pellet core, have been described in detail. Numerous commercially available formulations have also been presented. This reflects the use of the starter pellet as an excipient in a wide variety of active ingredients and dosage forms. It is important to note that no so-called universal standard exists among the various inert pellet cores that could be applied to all active ingredients or film coatings. For each drug technology development, the type and size fraction of the starter pellets must be selected according to the particular active ingredient, dosage strength, and dosage form. For the right decision, it is essential to have as much knowledge as possible about inert pellet cores.

## Figures and Tables

**Figure 1 pharmaceutics-14-01299-f001:**
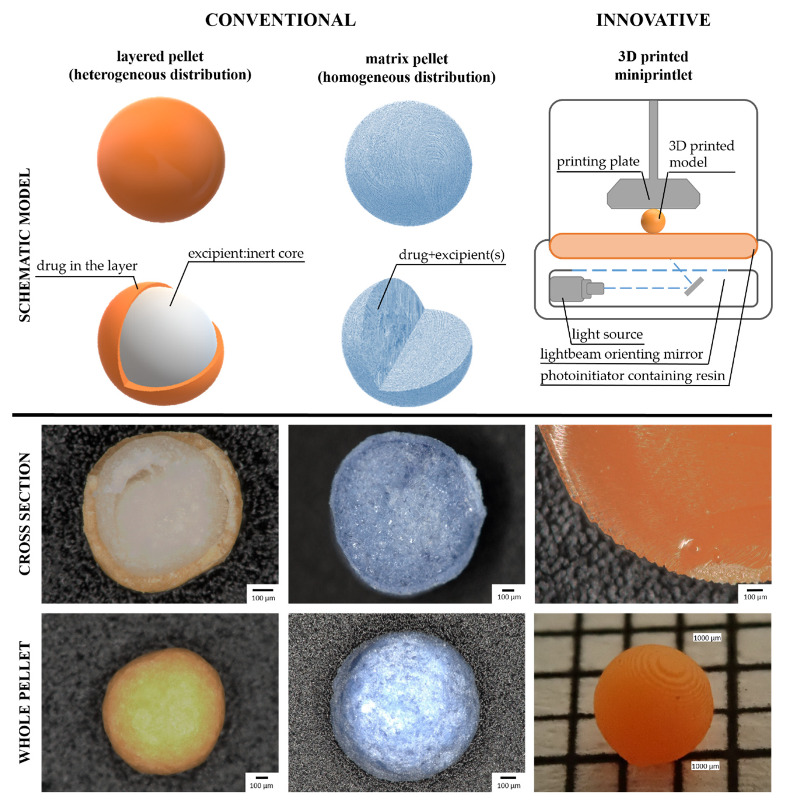
Pellets prepared by conventional and innovative technologies (Photos: Keyence VHX 970 digital microscope, Olympus Stylus TG-4 digital camera; 3D printer: Original Prusa SL1S SPEED 3D printer).

**Figure 2 pharmaceutics-14-01299-f002:**
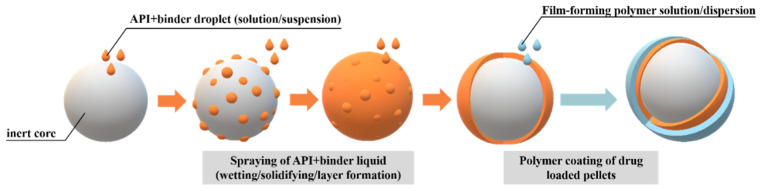
Schematic illustration of drug layering process of the active substance from solution/suspension and polymer coating.

**Figure 3 pharmaceutics-14-01299-f003:**
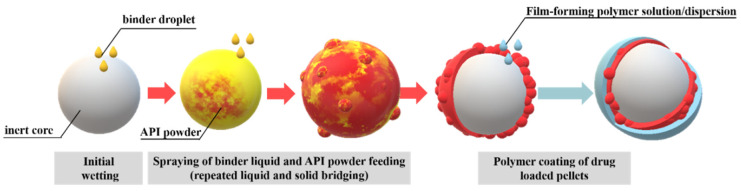
Schematic illustration of dry powder layering of starter core and polymer coating.

**Figure 4 pharmaceutics-14-01299-f004:**
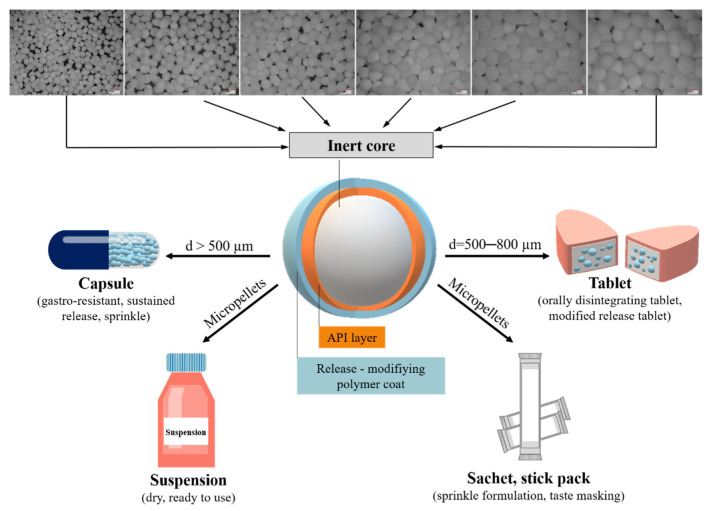
Relationship between particle size and dosage forms (Photo: Keyence VHX 970 digital microscope; inert core: sugar spheres of various particle sizes).

**Figure 6 pharmaceutics-14-01299-f006:**
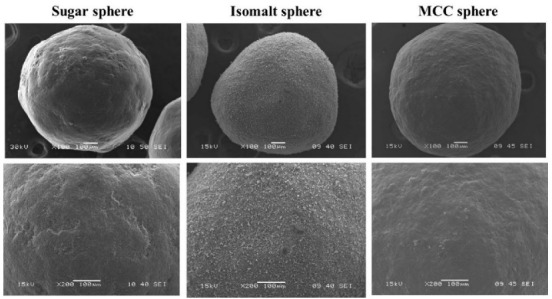
Scanning electron microscopic images of various inert cores (JEOL JSM-6380 LA).

**Figure 7 pharmaceutics-14-01299-f007:**
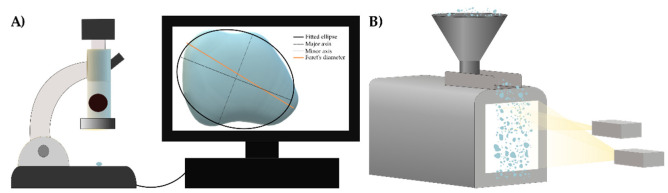
Schematic illustration of image acquisition methods ((**A**): static; (**B**): dynamic image analysis).

**Figure 8 pharmaceutics-14-01299-f008:**
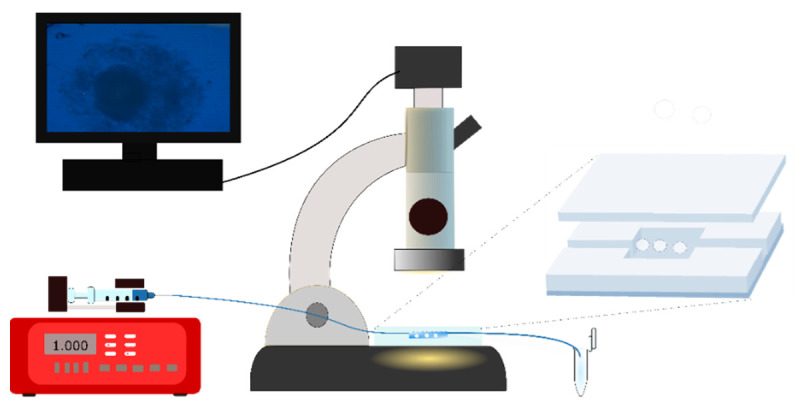
Experimental setup for pellet disintegration/dissolution test: (Microfluidic dissolution tester by Laki Technology (BioMicrofluidicsLab PPKE ITK), Pump (KF Technology, NE1000) Flow: 4000 µL/h, Nikon SMZ 1000 Optics 1×, Magn: 3×; NIS Elements Imaging Software).

**Figure 9 pharmaceutics-14-01299-f009:**
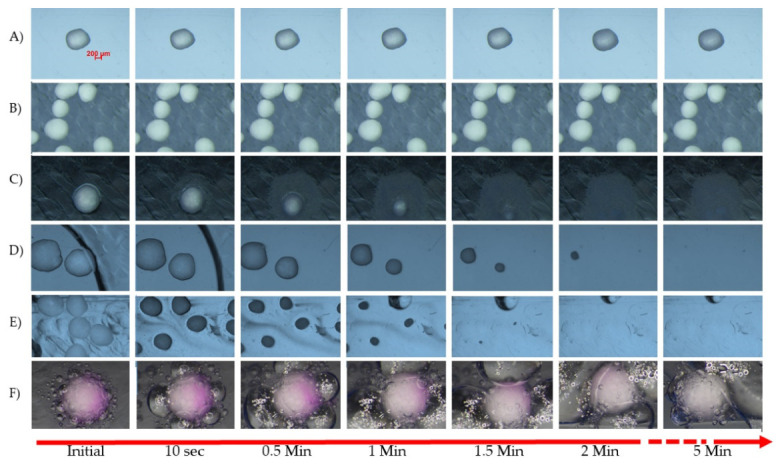
Effect of pH 1.2 HCl dissolution medium on different inert cores. (**A**) MCC spheres: Ethispheres^®^ (NPPharm Ltd., France); (**B**) DCPA spheres: PharSQ^®^ Spheres (Chemische Fabrik Budenheim, Budenheim, Germany; (**C**) Sugar core: Suglets^®^; (Colorcon, Bazainville, France); (**D**) Isomalt cores: galenIQ^TM^ 980 (Beneo GmbH, Mannheim, Germany); (**E**) Tartaric acid cores: TAP (IPC Process-Center GmbH, Dresden, Germany) (**F**) Sodium bicarbonate core: (Umang Pharmatech Pvt Ltd., Thane, India) (Latter medium contains Phenolptalein solution R (Ph. Eur. 10) as an indicator).

**Figure 10 pharmaceutics-14-01299-f010:**
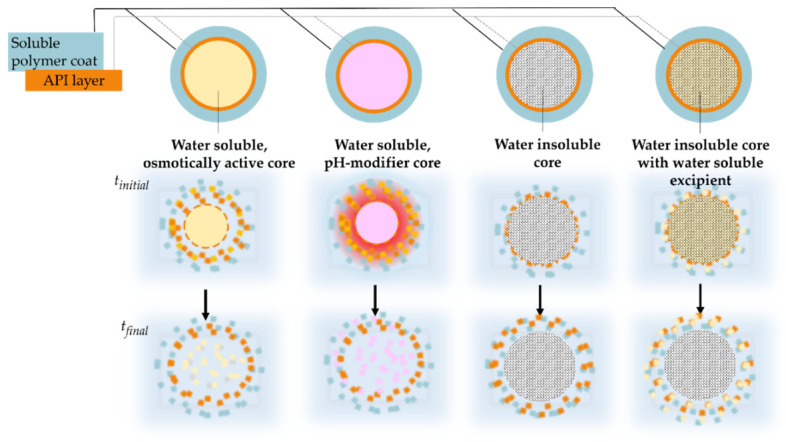
Schematic illustration of drug release from enteric-coated pellets (medium: high pH).

**Figure 11 pharmaceutics-14-01299-f011:**
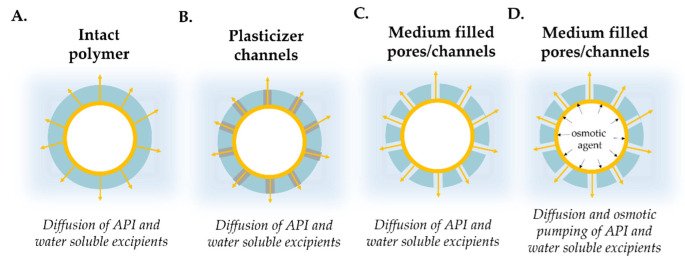
Schematic illustration of drug release mechanisms of coated pellets ((**A**): intact polymer; (**B**): polymer coat with plasticizer channels; (**C**): coat with medium filled pores/channels; (**D**): osmotically active core)).

**Figure 12 pharmaceutics-14-01299-f012:**
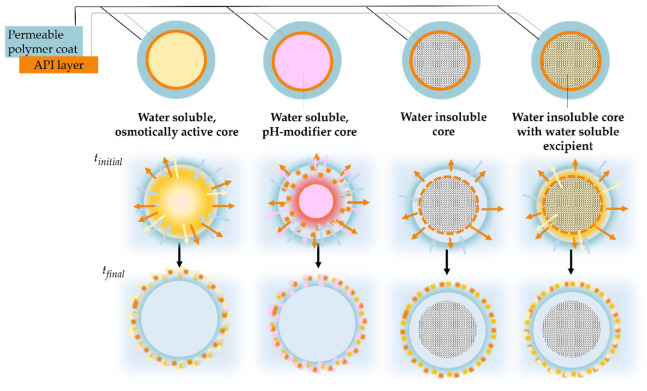
Schematic illustration of drug release from permeable coated pellets based on various starter cores.

**Figure 13 pharmaceutics-14-01299-f013:**
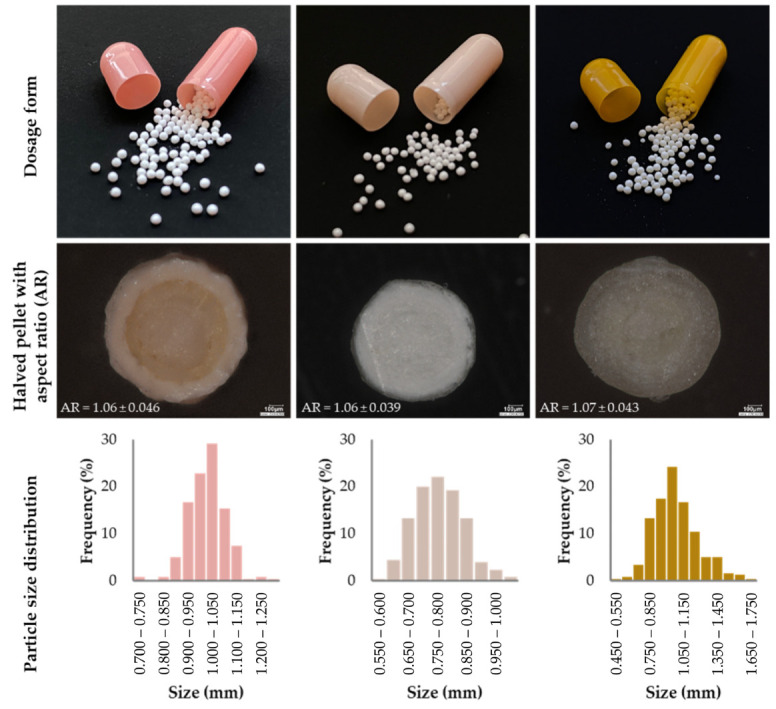
Microscopic photos and image analysis results (AR: aspect ratio; particle size distribution) of three commercially available products (1st column: Emozul^®^ (KRKA); 2nd column: Olwexya (KRKA); 3rd column: Cetebe (GSK); Keyence VHX 970 digital microscope; software: ImageJ; n = 200; Avg. ± SD evaluation of own experimental data).

**Table 1 pharmaceutics-14-01299-t001:** Commercially available starter spheres (based on [19,20] databases).

Pellet Core Material– Main Component	Brand Name (Manufacturer)
Sucrose	Suglets (Colorcon; Bazainville; France); Pharm-a-spheres (pharm-a-spheres GmbH; Tornesch; Germany); Surinerts (IPC Process-Center GmbH; Dresden; Germany); Vivapharm^®^ Sugar Spheres (JRS Pharma GmbH&Co.KG.; Weissenborn; Germany); Sugar pellets (Umang Pharmatech Pvt Ltd.; Thane; India)
Microcrystalline cellulose (MCC)	Vivapur^®^ MCC spheres (JRS Pharma GmbH&Co.KG.; Weissenborn; Germany); Cellets (IPC Process-Center GmbH; Dresden; Germany); CELPHERE™ (Asahi Kasei Corporation; Tokyo; Japan); MCC Pellets MS (Umang Pharmatech Pvt Ltd.; Thane; India)
Isomalt	galenIQ (Beneo GmbH; Mannheim; Germany)
Anhydrous dibasic calcium phosphate	PharSQ^®^ Spheres CM (Chemische Fabrik Budenheim; Budenheim; Germany)
Tartaric acid	TAP (IPC Process-Center GmbH; Dresden; Germany); Tartaric acid pellets (Umang Pharmatech Pvt Ltd.; Thane; India); InstaSpheres-TA (Tartaric acid spheres seal coated with hydrophilic polymer-Ideal Cures; Mumbai; India)
Silica	Silica Pellets AS (Umang Pharmatech Pvt Ltd.; Thane; India)
Xylitol	Xylinerts (IPC Process-Center GmbH; Dresden; Germany
Mannitol	Mannitol spheres SANAQ^®^ (Pharmatrans SANAQ; Basel; Switzerland)
Lactose	Lactose pellets LS (Umang Pharmatech Pvt Ltd.; Thane; India)
Calcium carbonate	Calcium carbonate SANAQ^®^ (Pharmatrans SANAQ; Basel; Switzerland); Calcium Carbonate Pellets CS (Umang Pharmatech Pvt Ltd.; Thane; India)
Starch	Starch Pellets STS (Umang Pharmatech Pvt Ltd.; Thane; India)

**Table 2 pharmaceutics-14-01299-t002:** Characteristic inert pellet core that can affect the quality of the final product.

Property	Consideration	Methods/Techniques/ Apparatus	Reference
**Particle size and particle size distribution**	final dosage form; processability (layering; blending), coating uniformity (drug load; polymer content)	sieve analysis; dynamic imaging analysis (DIA)	[54,55,56]
**Solubility**	processability, drug release	analytical measurement, image analysis	[48,57,58]
**Surface area**	processability (drug layer or coat thickness)	gas adsorption (BET); image analysis	[56]
**Surface roughness**	processability (drug layer or coat thickness)	image analysis 3D profilometer	[16]
**Tensile strength**	processability (fluidization, tableting)	Texture analyzer	[59]
**Friability**	processability	Ph.Eur. method	[60]

**Table 3 pharmaceutics-14-01299-t003:** Properties of some commonly used starter cores [2,48,58]. (+: present; -: not present).

	Sugar Spheres	Isomalt Spheres	MCC Spheres	TAP Spheres	Sealed TAP Spheres	DCPA Spheres
**Ingredient-Main component**	Sucrose	Isomalt	MCC	Tartaric acid	Tartaric acid	Dibasic calcium phosphate anhydrous
**Ingredient-additive(s)**	Maize starch	-	-	-	HPMC; PEG 400 Glycerol; MCC; Talcum	MCC
**Ingredients-official monograph (Ph.Eur./** **USP-NF)**	+	+	+	+	+	+
**Inert core-official monograph (Ph.Eur./** **USP-NF)**	+	-	-	-	-	-
**Water solubility**	*Sucrose:* soluble, *starch:* practically insoluble in cold water	Soluble	Insoluble	Soluble	TA: soluble; Coat: soluble/insoluble components	Insoluble

**Table 4 pharmaceutics-14-01299-t004:** Examples of sugar sphere-based formulations used in different dosage forms on the Hungarian market [144].

API	Dosage Form	Brand/Generic Names (Strength/mg)	Manufacturer
esomeprazole	Gastro-resistant hard capsule	Emozul (20; 40)	Krka d.d. (Novo mesto; Slovenia)
	Gastro-resistant tablet	Nexium (20; 40)	Astra Zeneca (Cambridge; UK)
omeprazole	Gastro-resistant hard capsule	Ludea (10; 20; 40)	Richter Gedeon Plc. (Budapest; Hungary)
lansoprazole	Gastro-resistant hard capsule	Lansacid (30)	TEVA Pharma S.L.U. (Alcobendas; Spain)
	Gastro-resistant hard capsule	Lansoptol (15; 30)	Krka d.d. (Novo mesto; Slovenia)
methylphenidate	MR hard capsule	Ritalin LA (20; 30; 40; 60)	Novartis (Basel; Switzerland)
naftidrofuryl	Retard capsule	Naftilong (100)	Hexal AG (Holzkirchen; Germany)
urapidil	Retard capsule	Ebrantil (30; 60; 90)	Altana (Wesel; Germany)
duloxetine	Gastro-resistant hard capsule	Cymbalta (30; 60)	Eli-Lilly S.A. (Indianapolis, Indiana; USA)
	Gastro-resistant hard capsule	Dulodet (30; 60)	Egis Plc (Budapest; Hungary)
tramadol	SR capsule	Adamon (50; 100; 150; 200)	Temmler Pharma GmbH (Marburg; Germany)
venlafaxine	Retard capsule	Olwexya (37.5; 75; 150)	Krka d.d. (Novo mesto; Slovenia)
mirtazapine	Orally disintegrating tablet (ODT)	Remeron (30; 45)	N.V. Organon (Oss; The Netherlands)
tizanidine	Retard capsule	Sirdalud MR (4; 6)	Novartis (Basel; Switzerland)
itraconazole	Retard capsule	Orungal (100)	Janssen-Cilag (Budapest; Hungary)
ascorbic acid	Retard capsule	Cetebe (500)	STADA Arzneimittel AG (Bad Vilbel; Germany)

**Table 5 pharmaceutics-14-01299-t005:** Examples of starter pellet formulations used in different dosage forms worldwide.

Dosage Form	Core Material	API	Brand Name	Manufacturer	Ref.
**Hard capsule**	MCC spheres 500	aprepitant	Aprepitant^®^ Sandoz	Sandoz (Basel; Switzerland)	[152]
**Gastroresistant hard capsule**	Tartaric acid core	dabigatran etexilate	Pradaxa^®^	Boehringer Ingelheim Pharmaceuticals (Ingelheim am Rhein; Germany)	[153]
**Prolonged-release hard capsule**	Sugar spheres	diltiazem hydrochloride	Dilcardia SR^®^	Mylan (Canonsburg, Pennsylvania; USA)	[154]
**Extended-release capsules**	Sugar spheres	memantine hydrochloride	Namenda XR^®^	Forest Laboratories Ireland LTD (Dublin; Ireland)	[155]
	Sugar spheres	dexmethylphenidate hydrochloride	Focalin XR^®^	Novartis (Basel; Switzerland)	[156]
	Sugar spheres	dextroamphetamine sulfate, dextroamphetamine saccharate, amphetamine aspartate monohydrate, amphetamine sulfate	Adderall XR^®^	Shire USA (Lexington, Massachusetts; USA)	[157]
	Sugar spheres	methylphenidate	Metadate CD^®^	UCB Inc. (Brussels, Belgium)	[158]
	Sugar starch spheres	morphine sulfate	Avinza^®^	King Pharmaceuticals R&D (Cary, Noth Carolina; USA)	[159]
	Sugar spheres	propranolol hydrochloride	Innopran XL^®^	ANI Pharmaceuticals (Baudette, Minnesota; USA)	[160]
**Oral capsules/Sprinkle capsules**	MCC Spheres 700	odevixibat	Bylvay^TM^	Albireo AB (Boston, Massachusetts;USA)	[161]
	Sugar spheres	topiramate	Topamax^®^ sprinkle capsule	Ortho-McNeil-Janssen Pharmaceuticals (Titusville, New Jersey; USA)	[162]
**Oral granules**	Sugar spheres	secnidazole	Solosec^®^	Catalent Pharma Solutions (Somerset; New Jersey; USA)	[163]
**Extended-release oral suspension**	MCC pellets	metformin hydrochloride	Riomet ER^®^	Sun Pharma (Goregaon, Mumbai; India)	[6]
**Gastro-resistant granules for oral suspension**	Sugar Spheres	esomeprazole Magnesium Trihydrate	Nexium^®^	Astra Zeneca (Cambridge; UK)	[164]
**Delayed-release oral suspension**	Sugar spheres	omeprazole magnesium	Prilosec^®^	Astra Zeneca (Cambridge; UK)	[143]
**Tablet multi-unit pellet system**	Sugar-starch pellets	omeprazole hemimagnesium	Antra^®^ MUPS	Cheplapharm Arzneimittel GmbH (Greifswald; Germany)	[8]
	Sugar spheres	omeprazole magnesium	Losec MUPS	Neon Healthcare (Hertford; UK)	[165,166]
**Orally disintegrating tablet (ODT)**	Lactose-MCC	lansoprazole	Prevacid Solutab Delayed-Release ODT	Takeda Pharmaceuticals (Tokyo; Japan)	[167]

## Data Availability

Not applicable.

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
