# Peer review of "Review on Starter Pellets: Inert and Functional Cores"

_pharmaceutics, 2022, doi:10.3390/pharmaceutics14061299_

Round 1
Reviewer 1 Report
The paper provides an overview of many available types and grades of the starter pellets available on the market, and examples of their application to formulate specific dosage forms. The fundamental mechanisms of the drug release for dosage forms started on those pellets are also provided. While there are also listed some key properties of the manufactured pellets usable as CQAs in QbD formulation development, any ambition to analyze the relationship between those properties and the formulation performance is lost quickly after beginning. The promise of analyzing specific of those properties as CQAs, which is present I the abstract of the paper is not fulfilled. Also, the graphical abstract shows rather the starter pellet usage, than the QbD and CQA aspects mentioned in the abstract
The cited references involve 145 items, of which 7 references are in Hungarian (incomprehensible for majority of readers), 14 are books, pharmacopoeia or another basic information, and 42 are prescribing info, product info or patent. That leaves 82 for “normal” papers. References 17, 33, 103 are non-specific and the exact referred document is hard to identify.
The paper certainly can be made as a link-hub to different pellet products on the market and examples of their application, but it does not provide much in terms of general conclusions and recommendations about how to select appropriate product from the portfolio for specific use. The description of the key properties of the pellets is on the textbook level, offering not much application-related evaluation.
As such, although it is well written, lacks ambitions to provide something more than an overview and I think it has not sufficient merit to be published in Pharmaceutics.
Some small comments on the specific details are as follows:
Figure 5 plots the particle size data, but there is no source of the data provided. Are those taken from product specifications or experimentally measured?
- 260 – A list of shape factors to characterize mostly spherical objects should include sphericity as a 3D property. Circularity and such are only simplifications used in connection with microscopy projection techniques.
Eqn. 5 – The source of coefficients in the equation should be cited at place, where f-factor is mentioned.
- 379 – MMC vs MCC
Author Response
First of all, the authors would like to thank the Reviewer for taking the time, and contributing their knowledge to improve their manuscript. We would like to thank Reviewer 1 for all the valuable comments and remarks.

Reviewer 2 Report
In the manuscript of Kállai-Szabó et al., the authors have reviewed the nature of the starter pellets in the finished multi-units dosage forms. The authors have described many possible inert pellets already available in the market and how the active product ingredient can be incorporated to the surface. Many examples of marketed, patented or under research pellets included in drug formulations are described in the manuscript. It is well written, and the bibliography is actual.
Author Response
The authors would like to express their graitude to the reviewer for taking the time to review the manuscript. Furthermore, your positive attitude and the praise you gave to our manuscript is highly appreciated. We hope to continue receiving your support.
Reviewer 3 Report
In this review manuscript, authors have highlighted the of various starter core of pellet. However, the scientific basis of the paper is poor (lacking the in-depth discussion).Therefore, this paper needs revision.
Comments
- The statement of line 48-51 requires the references.
- In line 379, please write the full form of “MMC”
- No need to write the abbreviated form of Critical quality attributes (CQAs) and Quality Target Product Profiles (QTPP) as the abbreviation is not used more than 1 times in the whole manuscripts.
- I recommend the authors to discuss on the critical process and formulation parameters such as inlet air temperature, m atomizing pressure, spraying rate, curing temperature that can affect the critical quality attributes of coated pellets (drug release rate, surface texture of coated pellets,, etc.) This is important because, various types of polymers (water soluble, water insoluble), coating process (coating using organic solvent and aqueous coating) on the basis of desired purpose of coating (taste masking, to protect the moisture sensitive drug, controlled release rate of the drug) is affect by the process and formulation factors.
- I recommend the authors to add the recent advance and future prospective of starter core pellets in oral drug delivery systems
Author Response
First of all, the authors would like to thank the reviewer for taking the time and contributing their knowledge to improve their manuscript. We would like to thank Reviewer 3 for all the valuable comments and remarks.

Reviewer 4 Report
The present review article focuses on starter pellets. It was a well written study, well structured, with plenty well presented schematic illustrations and figures. Also, the tables used in the review were very informative.
The recommendation is acceptance in present form.
Author Response
The authors are grateful to the reviewer for taking the time to review our manuscript. Furthermore, we highly appreciate the positive attitude and praise you gave to our manuscript. We hope to continue receiving your support.
Round 2
Reviewer 1 Report
I believe, the revised version of the manuscript removed all minor problems with the paper presentation and it also correctly redefined the paper objective in the abstract, so that it corresponds to the real content and the conslusions match both those objectives and the presented data. The list of references was improved and the rationale of reference selection was explained.
In the present form, the paper strength is in providing a link-hub to different pellet products on the market and examples of their application, which may be valuable for some groups of readers.
Some general conclusions and recommendations about how to select appropriate product from the portfolio for specific use is now provided as the description of key properties of the pellets is enhanced. Thus, while it is not perfect, it is no longer a weakness and it provides some more merit that justifies publication in Pharmaceutics.
Reviewer 3 Report
I appreciate the author’s responses and the revised manuscript is more improved than previous version. Therefore, I recommend this manuscript for publication.